# Comparative Cytological and Gene Expression Analysis Reveals That a Common Wild Rice Inbred Line Showed Stronger Drought Tolerance Compared with the Cultivar Rice

**DOI:** 10.3390/ijms25137134

**Published:** 2024-06-28

**Authors:** Zijuan Huang, Peishan Huang, Shihui Chen, Mengzhu Hu, Hang Yu, Haibin Guo, Muhammad Qasim Shahid, Xiangdong Liu, Jinwen Wu

**Affiliations:** 1State Key Laboratory for Conservation and Utilization of Subtropical Agro-Bioresources, South China Agricultural University, Guangzhou 510642, China; zjhuang@stu.scau.edu.cn (Z.H.); peishanhuang@stu.scau.edu.cn (P.H.); shchen@stu.scau.edu.cn (S.C.); mzhu@stu.scau.edu.cn (M.H.); hy@stu.scau.edu.cn (H.Y.); ghb@scau.edu.cn (H.G.); qasim@scau.edu.cn (M.Q.S.); 2Guangdong Provincial Key Laboratory of Plant Molecular Breeding, South China Agricultural University, Guangzhou 510642, China; 3Guangdong Base Bank for Lingnan Rice Germplasm Resources, College of Agriculture, South China Agricultural University, Guangzhou 510642, China

**Keywords:** *Oryza rufipogon* Griff, drought stress, seed germination, genome re-sequencing

## Abstract

Common wild rice (*Oryza rufipogon* Griff.) is an important germplasm resource containing valuable genes. Our previous analysis reported a stable wild rice inbred line, Huaye3, which derives from the common wild rice of Guangdong Province. However, there was no information about its drought tolerance ability. Here, we assessed the germination characteristics and seedling growth between the Dawennuo and Huaye3 under five concentrations of PEG6000 treatment (0, 5%, 10%, 15%, and 20%). Huaye3 showed a stronger drought tolerance ability, and its seed germination rate still reached more than 52.50% compared with Dawennuo, which was only 25.83% under the 20% PEG6000 treatment. Cytological observations between the Dawennuo and Huaye3 indicated the root tip elongation zone and buds of Huaye3 were less affected by the PEG6000 treatment, resulting in a lower percentage of abnormalities of cortical cells, stele, and shrinkage of epidermal cells. Using the re-sequencing analysis, we detected 13,909 genes that existed in the genetic variation compared with Dawennuo. Of these genes, 39 were annotated as drought stress-related genes and their variance existed in the CDS region. Our study proved the strong drought stress tolerance ability of Huaye3, which provides the theoretical basis for the drought resistance germplasm selection in rice.

## 1. Introduction

Rice (*Oryza sativa* L.) is a major food for more than 50% of the global population. The extreme weather caused by global climate change has seriously affected rice yield [1]. Drought stress is one of the influencing factors caused by extreme weather [2]. Drought stress disrupts the metabolism of rice tissues, resulting in a range of phenotypic, physiological, biochemical, and molecular level changes [3,4]. The drought tolerance capacity is essential to improve productivity under the drought stress condition. Drought tolerance is a complex trait involving rice germination, leaf rolling, tillering number, and root system development at different stages [5,6,7]. Understanding the mechanism of drought stress, especially for drought tolerance, will help to improve crop productivity under drought stress.

Seed germination is always coupled with morphological, physiological, and related enzyme changes. Drought stress negatively influences the germination process by disturbing the water balance, damaging the metabolic process at the cell level, and decreasing the amount of ATP production and respiration [8,9,10]. The germination time and germination rate will also be prolonged and slowed during the germination process under drought stress [11]. Therefore, the abilities of germinated seeds under drought stress indicates their drought resistance to some extent. For rice breeding scientists, improving seed germination under drought stress, such as identifying the drought-resistance genes or utilizing the potential germplasm resource, can increase crop yield.

Common wild rice (*Oryza rufipogon* Griff.) is a valuable germplasm resource and it is considered to be the ancestor of Asian cultivated rice (*Oryza sativa* L.) [12]. Compared with cultivar rice, wild rice has stronger resistance to environmental stress and virus resistance [13,14,15]. These characteristics are crucial for understanding and improving the stress tolerance of cultivated rice. To utilize the potential tolerance genes in wild rice resources, conventional hybridization, embryo rescue, embryo culture, protoplast fusion, and heterologous additional lines are frequently used in wild rice [16,17]. The wild rice genetic populations, such as inbred lines, introgression lines, and additional lines, are frequently used to identify the candidate resistance genes [14,18]. Many resistant genes, including insect resistance, cold tolerance, and drought tolerance genes, have been found in wild rice [19,20,21]. Two drought-resistance quantitative trait loci (QTLs), named the *qSDT2-1* and *qSDT12-2*, are detected by the introgression line of Dongxiang wild rice [18].

High-throughput sequencing technology, such as re-sequencing and RNA-seq analysis, plays a vital role in detecting genetic variation and differentially expressed genes (DEGs). Re-sequencing analysis was performed to identify the sequence polymorphisms, including the single-nucleotide polymorphisms (SNPs) and insertions/deletions (InDels), to gain new insights into different genome variations among the other rice varieties. The number of sequence polymorphisms, including the single-nucleotide polymorphisms (SNPs) and insertions/deletions (InDels), can be detected by the whole-genome re-sequencing analysis [22,23,24]. Comparative transcriptome analysis was performed to identify novel transcripts and gain new insights into different gene expressions and pathways during the rice development process. Under drought stress, over 5000 up-regulated and 6000 down-regulated DEGs were detected as differentially expressed in the rice [25,26]. These DEGs were divided into membrane transport, signaling, and transcriptional-regulated genes [27]. In the root tissue of perennial *O. rufipogon*, 37 candidate drought-resistant genes were identified under drought stress [1]. The ABA-dependent and ABA-independent regulation pathways were two primary regulatory pathways verified to play an important role in drought resistance mechanisms [28,29,30].

In a previous analysis, we constructed a stable wild rice inbred line, Huaye3, which was derived from the common wild rice of Guangdong Province. Huaye3 showed more robust resistance to rice blast and brown planthopper [31]. However, no information exists on its abiotic stress effect, especially under drought stress. Here, we used Huaye3 (drought-tolerant) and Dawennuo (drought-sensitive) as the materials. We compared their seed germination characteristics under the different gradients of PEG6000 solutions during the seed germination and seedling development stages. Then, WE-CLSM and semi-section analysis were used to analyze the root tip cell morphology under the different concentrations of PEG6000. Finally, genomic resequencing and gene expression analysis were conducted to verify the gene expression level of drought stress-related genes. Our study provides evidence of drought tolerance in Huaye3, and this study will facilitate the utility of these variations and resistance alleles in molecular breeding and functional genomics.

## 2. Results

### 2.1. Phenotypic Variation of Seed Germination in Huaye3 and Dawennuo under Drought Stress

To evaluate the germination differences between Huaye3 and Dawennuo, seed germination charts were investigated and summarized in this study (Figure 1, Figure 2 and Appendix A). Huaye3 showed a stable seed germination rate, with its value ranging from 98.33 ± 1.67% to 99.17 ± 0.83%, while the PEG6000 concentration was 15% and below (Figure 2A and Appendix A). The seed germination rates of Huaye3 still reached more than 52.50 ± 2.50% under the 20% PEG6000 treatment, while the seed germination rate in Dawennuo is only 25.83 ± 1.67% (Figure 2A).

We determined the seed relative vigor index of two materials under the drought stress treatment (Appendix A). With the increase of the PEG6000 concentration, the seed relative vigor index in Huaye3 and Dawennuo showed a similar tendency (Figure 2B). The seed relative vigor index of Huaye3 was much higher than Dawennuo at the 5% and 10% PEG6000 concentrations. The seed relative vigor index in Huaye3 under 5% and 10% PEG6000 ranged from 69.00 ± 1.26 and 58.72 ± 5.66, relatively, compared with 35.83 ± 2.55 and 26.25 ± 0.52 in Dawennuo. The drought tolerance index was also investigated in this study (Appendix A). The drought tolerance index of Huaye3 and Dawennuo decreased with the increasing treatment concentration (Figure 2C). The drought tolerance index in Huaye3 was higher than Dawennuo’s in each concentration treatment. The drought tolerance index of Huaye3 in 5%, 10%, and 15% PEG6000 concentration was 0.94 ± 0.04, 0.83 ± 0.01, and 0.68 ± 0.04, while compared with 0.86 ± 0.02, 0.77 ± 0.04, and 0.66 ± 0.04 in Dawennuo, relatively. These results further indicated that Huaye3 has a certain degree of drought tolerance.

### 2.2. Effects of Drought Stress on Seedling Stage between the Huaye3 and Dawennuo

To evaluate the growth variation of Huaye3 and Dawennuo in the seedling stage under the drought stress treatment, root number and fresh weight were selected and investigated Figure 3 and Appendix A). In this study, we investigated the growth of Huaye3 and Dawennuo in the seedling stage under drought stress on the 30th day (Figure 3A–D). The root number in Huaye3 showed no difference between the control and treatment groups (Figure 3E). The root number of Huaye3 ranged from 5.60 ± 0.25 to 6.67 ± 0.33 under the 15% PEG6000 and below. Comparatively, the average root number in Dawennuo was 14.00 ± 0.58 in the control group. However, the root number decreased to 6.67 ± 0.88 under the 15% PEG6000 treatment (Figure 3E). The results indicated that Huaye3 showed more robust stability under the PEG6000 treatment.

The fresh weight of seedlings in Huaye3 and Dawennuo under different concentrations of PEG6000 were further analyzed. With the rice seedlings of Huaye3 and Dawennuo grown for up to 30 days, we found both the Huaye 3 and Dawennuo were affected by the increase in PEG6000 concentration. Therefore, we selected five representative plants in each gradient treatment to evaluate the fresh weight of two materials. Compared with the no-treatment group, the fresh weight in Huaye3 showed a slight effect on the drought stress treatment, decreasing by 10.14 ± 2.19 mg and 20.88 ± 1.86 mg under the 10% and 15% PEG6000 treatments, respectively (Figure 3F). Compared with the no-treatment group in Dawennuo, the fresh weight was decreased by 18.30 ± 5.66 mg, 50.48 ± 5.72 mg, and 91.12 ± 5.64 mg under the concentration treatment of 5%, 10%, and 15% PEG6000, respectively (Figure 3F). The results indicated that the PEG6000 treatment showed less impact on Huaye3 than on Dawennuo.

### 2.3. Effects of Drought Stress on Root/Shoot Ratio and Moisture Content between the Huaye3 and Dawennuo

The root/shoot ratio and moisture content were also conducted to evaluate the growth variation of Huaye3 and Dawennuo in the seedling stage (Figure 4). The root/shoot ratio of Huaye3 in control was 0.68 ± 0.01, and it increased by 0.55 ± 0.05, 0.32 ± 0.01, 0.48 ± 0.04, and 1.12 ± 0.13, respectively, while under the PEG6000 concentration treatment of 5%, 10%, 15%, and 20% (Figure 4B). However, the root/shoot ratio value of Dawennuo in control is 0.46 ± 0.01, and it increased by 0.13 ± 0.01, 0.34 ± 0.04, and 0.60 ± 0.06, respectively, under the PEG6000 concentration treatment of 5%, 10%, and 15% (Figure 4B). In addition, the root/shoot ratio of Dawenuo decreased by 0.07 compared to the control at 20% PEG6000 concentration treatment.

The moisture content of seedlings was also investigated in this study. With the increase of the PEG6000 concentration, the moisture content in Huaye3 and Dawennuo decreased. The moisture content of Huaye3 in the control was 77.53 ± 0.51 and decreased by 5.58 ± 0.49, 8.26 ± 0.79, 14.14 ± 0.68, and 25.30 ± 3.15 under the PEG6000 concentration treatment of 5%, 10%, 15%, and 20%, respectively (Figure 4C). The moisture content of Dawennuo in the control was 77.64 ± 0.82 and decreased by 2.18 ± 1.66, 8.93 ± 2.21, 19.38 ± 1.77, and 42.86 ± 4.38 under the PEG6000 concentration treatment of 5%, 10%, 15%, and 20%, respectively (Figure 4C). These results indicated that Huaye3 showed a lower decrease in the moisture content of seedlings.

### 2.4. Effects of Drought Stress on Root Cell Structure between the Huaye3 and Dawennuo

To reveal the cause of high drought tolerance in Huaye3, the cellular structure of the root tip elongation zone on the 7th germination day was observed (Figure 5). In the Dawennuo, the root tip elongation zone cells in control were intact and neatly arranged (Figure 5A,A1). With the PEG6000 concentration increasing to more than 5%, the epidermis, cortex, and stele region cells appeared abnormal. Under the 5% and 10% PEG6000 treatment, the cortical cell layers in the apical elongation zone were increased (Figure 5C,E). In the stele, we detected that the stele area was more significant than before, and the cellulose and vessel in the stele were also increased (Figure 5C1,E1). In addition, we detected that epidermal cells were thickened under the 10% concentration treatment (Figure 5E). The PEG6000 concentration increased to 15%, and the root tissue, epidermis, cortex, and sclerenchyma cells were shriveled and deformed (Figure 5G). Notably, the stele was deformed and irregular (Figure 5G1). When the PEG6000 concentration increased to 20%, the root tissue had much more severe deformation and serious change (Figure 5I).

Compared with Dawennuo, the root tip elongation zone structure in Huaye3 exhibited no difference under the 5% and 10% concentrations of PEG6000 treatment (Figure 5B,B1,D,D1,F,F1). When the PEG6000 concentration increased to 15%, the cortical and stele of the root tip elongation zone in Huaye3 became abnormal (Figure 5H,H1). Compared with the control, the smaller size of cortical cells and increased cortical layers were detected in Huaye3 at the 15% and 20% PEG6000 concentrations (Figure 5H,J). Notably, several shrinkages of epidermal cells and irregular deformation of stele cells were detected in the Huaye3 under the 20% PEG6000 concentration (Figure 5J,J1). These results showed that the root system of Huaye3 was less affected by PEG6000 drought stress than Dawennuo.

We further observed the root tip elongation zone of Huaye3 and Dawennuo using the WE-CLSM analysis (Figure 6). The root tip elongation zone cells of Dawennuo in the control group were uniform in size and neatly arranged (Figure 6A,A1). With the PEG6000 concentration increased to 5%, the number of cortical cell layers increased compared with the control (Figure 6C,C1). When the concentration of PEG6000 reached 10% and above, we found significant changes in the cortex cells compared to the control. The cortical cells in the root tip elongation zone of Dawennuo were deformed and irregularly shaped at the 10% PEG6000 concentrations (white arrow) (Figure 6E,E1). Notably, the cortical cells in the root tip elongation zone of Dawennuo were more disorganized, with severe cellular aberrations and apparent irregular shape at 15% and 20% PEG6000 concentration (Figure 6G,G1,I,I1).

We also observed the root tip elongation zone cells of Huaye3 and found that the cells in the root tip elongation zone were clearly and neatly arranged in the control (Figure 6B,B1). Compared with the control, there was no significant differences in the root tip elongation zone cells in 5% PEG6000 concentration (Figure 6D,D1). When the PEG6000 concentration reached 10% and above, the cell arrangement of the root tip elongation zone cells of Huaye3 was still precise and neatly arranged. In addition, the number of epidermal cell layers increased (Figure 6F,F1,H,H1,J,J1). Thickening of epidermal cells in the root tip elongation zone cells occurred at a PEG6000 concentration of 20% (Figure 6J,J1). These results suggested that drought stress affected the number of cortical cells and cortex in the root tip elongation zone. The results also showed that the root system of Huaye3 was less affected by drought stress than Dawenuo.

### 2.5. Effects of Drought Stress on Rice Bud Cells between the Huaye3 and Dawennuo

To evaluate the drought stress effects on rice bud development between the two materials, we observed the bud cellular structure under the different concentrations of PEG6000 on the 7th germination day. In the control of Dawennuo and Huaye3, the bud’s cellular structure comprised a developing incomplete leaf and two complete leaves (Figure 7A,B). Large air cavity tissues and clearer vein cells were easily observed in the developing, incomplete, and complete leaves (Figure 7A1,B1). The outermost germinal sheath of Dawennuo develops normally and sheds normally (Figure 7A).

With the PEG6000 concentration increased to 5%, two pieces of complete leaves have been formed in the Dawennuo and Huaye3, respectively (Figure 7C,D). The cell structures of leaves were detected in two materials (Figure 7C1,D1). With the PEG6000 concentration increased to 10%, we detected that the bud structure cells of the two materials had a significant difference. In Dawennuo, the first piece of the leaf could not develop clear vein cell structures and we could not find the second complete leaf (Figure 7E,E1). Compared with Dawennuo, we observed two pieces of growing leaves, and the leaf cellular structure and vascular tissue in Huaye3 were usually developed (Figure 7F,F1).

With the PEG6000 concentration increasing to 15%, we observed the first complete leaf in the Dawenuo and Huaye3, respectively (Figure 7G,H). The air cavities of the incomplete leaf were significantly smaller than their respective controls (Figure 7G1,H1). With a concentration of up to 20% PEG6000, we could not observe the developing leaf in these two materials. Only coleoptiles were detected (Figure 7I,J). In addition, the bulliform cells in the coleoptile of Huaye3 appeared under the 20% concentration of PEG6000 treatment (Figure 7J1).

These results indicated that drought stress also affected the bud development of Huaye3 and Dawennuo. However, the bud development of Huaye3 was less affected by drought stress than Dawenuo’s.

### 2.6. Re-Sequencing Analysis Revealed Significant Variations in Huaye3

To reveal the cause of strong tolerance ability in Huaye3 under drought stress, we detected the differences in single nucleotide polymorphisms (SNPs) and insertions/deletions (InDels) between Huaye3 and Dawennuo at the genome level (Figure 8). The results showed that Huaye3 had the highest number of single nucleotide polymorphisms (SNPs) on chromosome 1 and the least number of SNPs variants detected on chromosome 12 compared with the Dawennuo (Figure 8A). In terms of the InDels, Huaye3 detected the highest number of InDels on chromosome 1 and the lowest number of InDels on chromosome 12 compared with Dawennuo (Figure 8B).

We focused on the two types of genetic variation in Huaye3, including the non-synonymous SNP mutations and InDels of the CDS region. These variations may affect the gene expression level of the Huaye3. A total of 311,628 SNPs involved in 37,267 genes were predicated on the existence of differences in Huaye3 (Figure 8C). These genetic variations primarily belong to non-synonymous mutations, resulting in gene coding region variations. In addition, two types of InDels, which contained insertions and deletions, were also detected and analyzed in this study. A total of 40,156 InDels involved in 15,609 genes were detected in Huaye3 (Figure 8C). These results indicated that 13,909 genes existed the significant differences in Huaye3 compared with the cultivar rice (Figure 9A, Appendix A).

GO enrichment analysis was used to further analyze these differentially expressed genes (DEGs). Three function categories were detected, including the biological process, cell component, and molecular function, all of which showed significant variation in Huaye3 (Figure 9B–E). In the biological process term, DEGs were mainly over-represented in the DNA metabolic process, RNA-dependent DNA replication, DNA replication, chromatin assembly or disassembly, defense response, and stress response process (Figure 9B,E). In the cell component term, the DEGs were over-represented primarily on chromatin, chromosome, intracellular organelle part, non-membrane-bounded organelle, and nucleus (Figure 9C,E). In the molecular function term, the DEGs were mainly over-represented in nucleic acid binding, DNA polymerase activity, DNA binding, and chromatin binding (Figure 9D,E). These results indicated that Huaye3 showed a significant difference compared with Dawennuo.

### 2.7. Drought-Related Genes Were the Primary Up-Regulation in Huaye3 Compared with Cultivar Rice

We integrated Huaye3’s genomic variation with known drought sensitivity and drought stress resistance genes. Through this comparison, 39 drought stress-related genes were found to be different in the CDS region of Huaye3 (Appendix A). Among these genes, several drought stress-related genes were detected and have been functionally analyzed, including *LOC_Os01g43650*, *LOC_Os05g15530*, *LOC_Os06g14030*, and *LOC_Os11g05160*, which encoded the drought-related proteins (Figure 10A). For example, *OsWRKY11* (*LOC_Os01g43650*) is a *WRKY* transcription factor, and its transgenic lines showed significant resistance to heat and drought, as well as low humidity resistance after the heat pretreatment [32]. *OsDIAT* (*LOC_Os05g15530*) encodes drought-induced branched-chain amino acid aminotransferase [33]. *OsGORK* (*LOC_Os06g14030*) encodes an outward-rectifying shaker-like potassium channel [34]. *OsAHL1* (*LOC_Os11g05160*) is an AT-hook motif, and PPC domain-containing protein and overexpression of *OsAHL1* enhanced the tolerance of rice seedlings to drought, salt, and cold stress and significantly improved the tolerance of rice to drought stress at the panicle development stage [35].

We selected four genes from this study in order to analyze their gene expression in root tissues after five days of 0% and 15% PEG6000 concentration using RT-qPCR analysis. The RT-qPCR results for *LOC_Os01g43650*, *LOC_Os05g15530*, *LOC_Os06g14030*, and *LOC_Os11g05160* are consistent with the re-sequencing analysis. These results indicated the reliability and accuracy of the re-sequencing analysis (Figure 10B–E).

## 3. Discussion

### 3.1. Huaye3 Showed Stronger Germination Characteristics Compared with the Cultivar Rice

Improving rice yield is urgently needed to meet the food shortage crisis. However, extreme weather, including drought stress, has seriously affected rice yield [2]. Drought stress is a complex factor affecting morphological, physiological, biochemical, and molecular processes. It can severely reduce rice grain yield, germination, growth rates, and tillering number [6,7]. Therefore, utilization of the potential drought-resistance germplasm and identification of the drought-resistance genes play an important role in rice breeding.

Common wild rice can survive in extreme conditions. It contains stress-resistant genes and shows greater stress resistance than cultivar rice [12,14,18]. Huaye3 is a stable wild rice inbred line developed by our lab, and it includes stronger resistance to rice blasts and brown planthopper [31]. Until now, there has been very little research about its drought tolerance in germination characteristics and seedling stage. Therefore, it is of great significance to explore and reveal the characteristics of Huaye3 under drought stress. PEG6000 is frequently used to evaluate drought stress in rice [36,37]. Rice germination was greatly affected by PEG6000 concentration under drought stress [38,39]. The critical PEG6000 concentration for drought tolerance in cultivar rice is 10–20%. Here, we used five different gradients of PEG6000 solutions to simulate the drought-stress environment. Compared with the drought-sensitive cultivar rice of Dawennuo, Huaye3 showed a higher seed germination rate than the cultivar rice. The germination rate of Huaye3 reached 52.50 ± 2.50% under the 20% PEG6000 treatment; for Dawennuo, it is only 25.83 ± 1.67%. In addition, the germination characteristics of the seed relative vigor index and drought tolerance index also showed that Huaye3 showed a stronger tolerance than the cultivar rice. The drought tolerance index in Huaye3 was higher than Dawennuo’s in each concentration treatment. These results indicated that Huaye3 showed a stronger drought tolerance than the Dawennuo.

The seedling stage is the other sensitive stage for plant development under drought stress. From this study, we investigated the variance of root number, fresh weight, root/shoot ratio, and seedling water content between the two materials. The root/shoot ratio is an important index for the drought tolerant ability. The root length, seedling length, and fresh weight were inhibited in varying concentrations of PEG6000 treatment. The root number of Huaye3 showed more stability and there was a slight effect on the fresh weight under the drought stress treatment when compared with Dawennuo. These results were similar to a previous study. Two rice varieties, Swarnaprabha and Kattamodan, showed strong drought recovery ability, and their root/shoot ratio increased under the high concentration of PEG6000 [40]. In addition, the fresh weight in Huaye3 showed a slight effect on the drought stress treatment compared to the control group in Dawennuo. These results indicate that Huaye3 is more stable than the Dawennuo under drought stress treatment.

### 3.2. Stable Adaptability in Root Tissue Is Probably the Primary Cause for the Drought Resistance in Huaye3

Drought stress is a critical environmental factor limiting plant growth and productivity [41]. Root tissue is generally responsive to the drought-stress environment. The phytohormones, such as the Strigolactones (SLs), regulate the root system and responsiveness to the drought stress treatment [42]. Under the drought stress treatment, the sclerenchyma lay cells and endodermis suberization in root tissue decreased and increased, respectively [43]. In addition, the secondary root growth and primary root structures were limited and thickened by the drought-stress environment [44,45]. Different anatomical structures and physiological modifications in root tissues were detected between the drought-sensitive and drought-tolerant varieties [27]. Therefore, we compared the cellular structure of the root tip elongation zone at varying concentrations of PEG6000 in this study.

Compared with the Dawennuo, Huaye3 showed a stronger drought resistance ability with the increased PEG6000 concentration and prolonged treatment time. The epidermal cell and stele are the primary structures for the root tissue, and they play an essential role in water absorption and transport. The smaller cortical cells and increased cortical layers were detected in Huaye3 under the different concentrations of PEG6000 treatment. From this study, we also observed the apical elongation zone of Dawennuo and found that cortical and stele cells were easily deformed and irregularly shaped under the 15% and 20% PEG6000 treatments. These results were similar to the rice material of Shanyou63 under drought stress. The root tissue of Shanyou63 under drought stress was significantly contracted and distorted compared with the no-treatment group [46]. In addition, our WE-CLSM results indicated that the stability in epidermal cells and outer skin cells of Huaye3 adapts to the drought-stress environment and improves the water absorption capacity of the root system. These results indicated that drought stress influenced the cortical cell and the number of cortical layers in the apical elongation zone. In addition, the results also demonstrated that the 15% and 20% PEG6000 are suitable treatment concentrations for drought stress evaluation and drought-resistant materials selection.

### 3.3. Huaye3 Existed the Significant Genetic Variation Compared with Cultivar Rice

The resequencing analysis is a valuable tool to detect the genetic variation of rice varieties. The single-nucleotide polymorphisms (SNPs) and insertions/deletions (InDels) were used to reveal new insights into sequence polymorphisms in rice. In a previous analysis, our lab constructed a stable wild rice inbred line, Huaye3, deriving from the common wild rice of Guangdong Province. Huaye3 showed a stronger resistance to rice blast and brown planthopper [31]. In the present work, we verified that Huaye3 showed stronger drought adaptability to drought stress. Therefore, we evaluated the genome-level variation between the Huaye3 and Dawennuo to confirm the relationship between genome-level variation and drought stress.

A total of 13,909 genes were detected in Huaye3 and existed in terms of genetic variation compared with Dawennuo. From this study, we combined the differentially expressed genes (DEGs) of Huaye3 with the drought stress-related genes. Drought stress-related genes are involved in the drought stress-responsive and stress-tolerant genes. Until now, many drought stress-related genes have been shown to be involved in plant adaptation to abiotic stresses [47,48,49,50]. Here, 39 drought-sensitive and drought-tolerance genes were detected, and the genetic variance existed in the CDS region. Moreover, we also verified the expression levels of four drought-stress-related genes, and these results showed the up-regulation of these genes under drought stress.

Several genes related to drought stress were identified during functional analysis, including *LOC_Os01g43650*, *LOC_Os05g15530*, *LOC_Os06g14030*, and *LOC_Os11g05160* [32,33,34,35]. These genes encode drought-related proteins, such as the *WRKY* transcription factor *OsWRKY11* (*LOC_Os01g43650*), drought-induced branched-chain amino acid aminotransferase *OsDIAT* (*LOC_Os05g15530*), outward-rectifying shaker-like potassium channel *OsGORK* (*LOC_Os06g14030*), and AT-hook motif and PPC domain-containing protein *OsAHL1* (*LOC_Os11g05160*). Overexpression of *OsAHL1* enhanced the tolerance of rice seedlings to drought, salt, and cold stress and improved the tolerance of rice to drought stress at the panicle development stage. The gene expression results indicated that these drought stress genes are up-regulated and highly expressed in the high concentrations of PEG6000, which could explain why Huaye3 has strong drought stress adaptability and drought tolerance. These results provide valuable insights into the regulatory mechanism of drought tolerance in Huaye3.

## 4. Materials and Methods

### 4.1. Plant Materials

Two materials, including the wild rice inbred line ‘Huaye3’ (drought-tolerant) and the cultivar rice ‘Dawennuo’ (drought-sensitive), were used in this study. The seeds of the two genotypes were harvested simultaneously and stored in the same environment. Seeds with full grains and consistent size were randomly selected and cleaned with distilled water after disinfection with 2.5% sodium hypochlorite for 30 min. Then, 40 seeds were placed in 9 cm diameter petri dishes covered with filter paper and treated with 0, 5%, 10%, 15%, and 20% PEG6000 solutions. Three biological replicates were performed in this study. The Petri dish was placed in a 26 °C incubator, and the PEG6000 solution was checked and renewed every two days.

### 4.2. Measurement of Seed Germination Characteristics

Three germination traits were selected and used in this study to evaluate the variance of seed germination under the different PEG6000 concentrations. Among these seed germination traits, the germination standard of rice seeds indicated 1 mm of radicle breakthrough and a germ length of 1/2 of the seed length [51]. The seed’s germination rate under the different treatments was investigated from the 3rd day. The relative vigor index and drought tolerance index were measured according to a previous study [52,53]. The relative vigor index (RVI) was calculated as = (VI of treatment group/VI of the control group) × 100, Vigor index (VI) was calculated as = Germination index (GI) × S (S: Total length of roots and shoots of seeds). The drought tolerance index was calculated as = GI of the treatment group/GI of the control group [52,53]. All these samples were subjected to three replications.

### 4.3. Measurement of the Morphological Index in the Seedling Stage in Two Materials

On the 8th day, the germinated seeds were selected and transferred into a culture hydroponic box. All culture hydroponic boxes were added and treated with the corresponding concentrations of PEG6000 solution. Four seedling stage indexes were used to measure the seedling under the drought stress treatment, including the total number of roots, seedling fresh weight, root shoot ratio, and moisture content seedlings [54]. Five representative plants were selected and evaluated for all morphological indexes in the seedling stage.

### 4.4. Cytological Analysis of Root Tip and Bud Tissue under the Drought Stress

To verify the root and bud structure difference under the drought stress treatment, we collected the root tip and bud samples from different concentrations of PEG6000. On the 7th day, the root and bud samples were collected and fixed in the FAA solution for 48 h. Then, samples were dehydrated with 50%, 70%, 80%, 90%, and 95% gradient ethanol. According to manufacturer instructions, the root tip was embedded in a Technovit 7100 embedding kit (Kulzer Technik, Germany). Embedded samples were further sectioned by the Leica RM2235 microtome and stained with 1% toluidine blue O [55].

WE-CLSM analysis was also conducted to observe the root samples of Huaye3 and Dawennuo. The collected samples were rehydrated by gradient ethanol and treated with the eosin B (C20H6N2O9Br2Na2, FW 624.1) solution. The detailed procedures had been described previously [56]. The samples were scanned and observed using a Leica SPE laser scanning confocal microscope (Leica Microsystems, Heidelberg, Germany) at a laser wavelength of 543 nm.

### 4.5. Re-Sequencing Analysis of Huaye3 and Dawennuo

To evaluate the genetic variation of Huaye3 and Dawennuo, the leaf samples of Huaye 3 and Dawennuo were collected and sent to Biomarker Technologies (Beijing, China) for genome resequencing analysis. The sequencing libraries were constructed by the HiSeqTM2500 platform (Illumina, USA) and sequenced according to the manufacturer’s instructions. Then, sequencing reads were aligned to the *Japonica* Nipponbare reference genome using BWA (0.7.17-r1188) software. Genome Analysis Toolkit (GATK, version 4.2.2.0) software tools were used to identify the polymorphic sites of SNPs and InDels. SNPs and InDel annotations were performed using SnpEff (4.3s) software. The agriGo and String datasets used the GO and protein-protein analyses, respectively [57,58].

### 4.6. RT-qPCR Analysis

The root samples were collected from Huaye3 and Dawennuo under the 0 and 15% PEG6000 treatments on the 5th day. Each sample contained three biological replicates and was stored and kept at −80 °C until RNA extraction. The AG TRIzol Reagent kit was used to extract the total RNA. The RNA quality was assessed by formaldehyde agarose gel electrophoresis and quantitated by spectrophotometer [59]. Then, cDNA was synthesized by reverse transcription experiments. Four representative drought-stress genes were selected for real-time PCR (RT-qPCR) analysis. The Primer3 Plus website designs the primers of targets, and their product length ranges from 60 to 150 bp (Appendix A). The specificity of the primer sequence was determined using the Primer Blast website in NCBI. The primer was synthesized by Sangon Bioengineering (Shanghai, China), and their sequences are listed in Appendix A. The RT-qPCR experiment was performed on the Roche Lightcycler480 instrument with the final volumes of 20 μL, each containing 1 μL of cDNA, 0.4 μL of each primer (10 μM), 10 μL 2× Hieff qPCR SYBR Green Master Mix (No Rox), and 8.2 μL ddH_2_O. The reaction conditions were as follows: 30 s at 95 °C, 40 cycles of 95 °C denaturation for 10 s, and 60 °C annealing and extension for 30 s. The rice ubiquitin gene was used as the reference gene, and its sequence is listed in Appendix A. The specificity of PCR results was verified by Melting curves analysis (Appendix A). After the reaction data were obtained, the gene expression level was calculated using the 2^−ΔΔCt^ method [60]. The RT-qPCR for each gene had three technical replicates and three biological replicates.

### 4.7. Statistical Analyses

Excel 2021 software was used to collect and analyze the drought stress-relevant data in the study. The statistically significant difference analysis was conducted using one-way ANOVA analysis using IBM SPSS Statistics 25.0 software. All results are presented as means ± standard derivation (SD) and have three biological replicates. GraphPad Prism 8.0 software was used for graphing.

## 5. Conclusions

In this study, we assessed the drought tolerance ability of Huaye3 and compared it with that of cultivar rice. The results showed that Huaye3 exhibited stronger drought tolerance, with a seed germination rate of more than 50%, even when exposed to a concentration of PEG6000 that exceeded 20%. We conducted cytological and resequencing analysis, which revealed that the roots and buds of Huaye3 were less affected by the PEG6000 treatment. We also identified 39 drought stress-related genes that existed in genetic variation between Huaye3 and the cultivar rice. The findings suggest that Huaye3 is a promising material for rice breeding and could potentially be used as a new drought-tolerant variety.

## Figures and Tables

**Figure 1 ijms-25-07134-f001:**
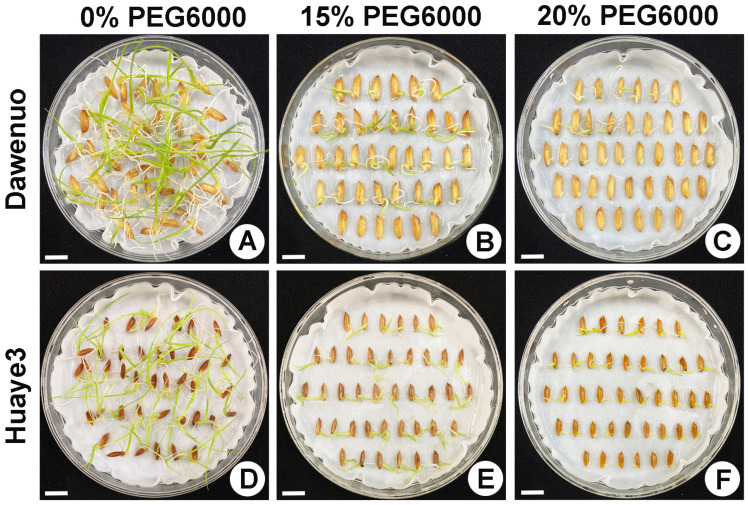
Comparison of seed germination in two materials under different concentrations of PEG6000 on the 7th day. (**A**) The seed germination of Dawennuo on the 7th day with no PEG6000, CK. (**B**) The seed germination of Dawennuo on the 7th day under 15% PEG6000. (**C**) The seed germination of Dawennuo on the 7th day under 20% PEG6000. (**D**) The seed germination of Huaye3 on the 7th day with no PEG6000, CK. (**E**) The seed germination of Huaye3 on the 7th day under 15% PEG6000. (**F**) The seed germination of Huaye3 on the 7th day under 20% PEG6000. Bars = 1 cm.

**Figure 2 ijms-25-07134-f002:**
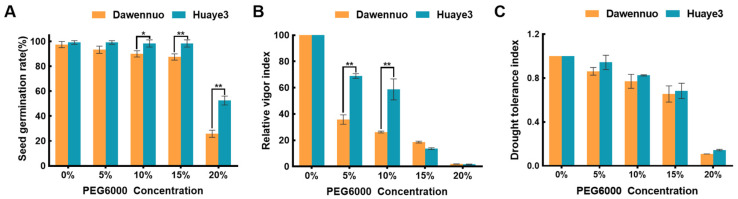
Comparison of germination characteristics under different concentrations of PEG6000. (**A**) Comparison of seed germination rate between the Dawennuo and Huaye3 under the different concentrations of PEG6000. (**B**) Comparison of seed relative vigor index between the Dawennuo and Huaye3 under the different concentrations of PEG6000. (**C**) Comparison of drought tolerance index between the Dawennuo and Huaye3 under the different concentrations. * and ** represent the significant and extremely significant differences when *p* < 0.05 and *p* < 0.01, respectively.

**Figure 3 ijms-25-07134-f003:**
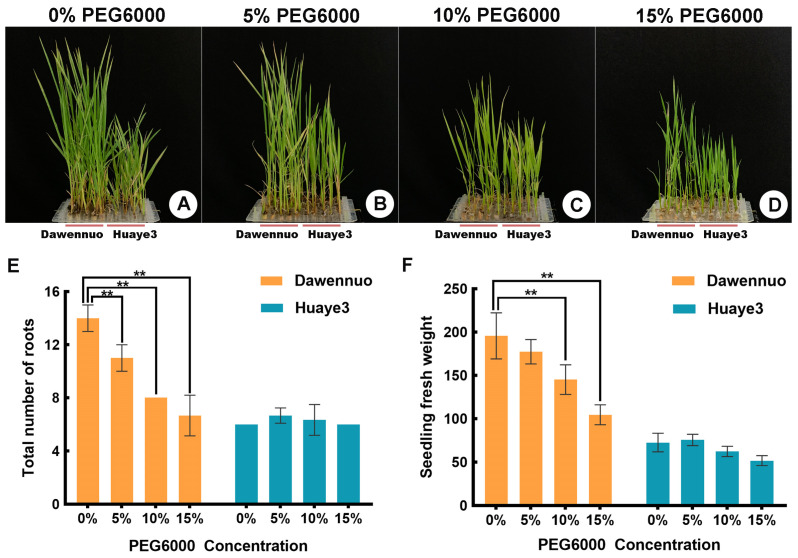
Comparison of Dawennuo and Huaye3 under different concentrations of PEG6000 at the seedlings stage. (**A**–**D**) Comparison of seedlings under the different concentrations of PEG6000. (**A**) Comparison of seedlings with no treatment in Dawennuo and Huaye3. (**B**) Comparison of seedlings between the Dawennuo and Huaye3 under the 5% PEG6000. (**C**) Comparison of seedlings between the Dawennuo and Huaye3 under the 10% PEG6000. (**D**) Comparison of seedlings between the Dawennuo and Huaye3 under the 15% PEG6000. (**E**) Comparison of the total number of roots between the Dawennuo and Huaye3 under the different concentrations of PEG6000. (**F**) Comparison of seedling fresh weight between the Dawennuo and Huaye3 under the different concentrations of PEG6000. ** represent the highly significant differences when *p* < 0.01.

**Figure 4 ijms-25-07134-f004:**
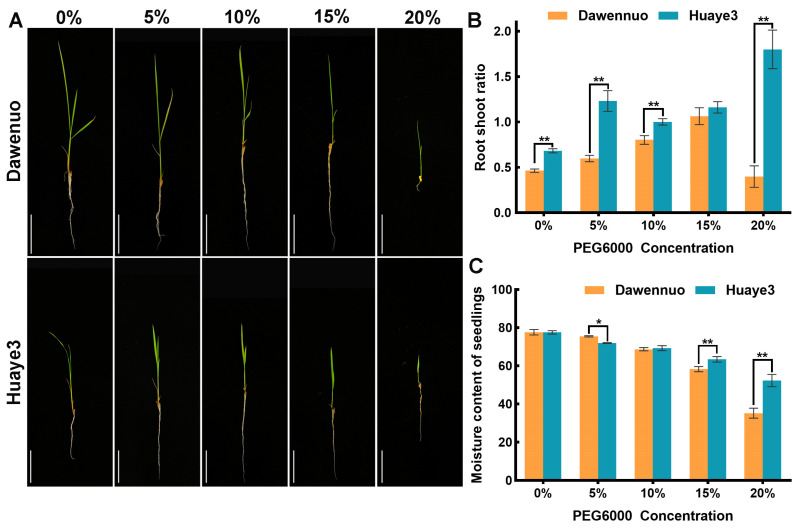
Comparison of root/shoot ratio and moisture content in Dawennuo and Huaye3 at the seedlings stage. (**A**) Comparison of seedlings between the Dawennuo and Huaye3 under the different concentrations of PEG6000. (**B**) Comparison of root/shoot ratio between the Dawennuo and Huaye3 under the different concentrations of PEG6000. (**C**) Comparison of moisture content of seedlings between the Dawennuo and Huaye3 under the different concentrations of PEG6000. * and ** represent the significant and extremely significant differences when *p* < 0.05 and *p* < 0.01, respectively.

**Figure 5 ijms-25-07134-f005:**
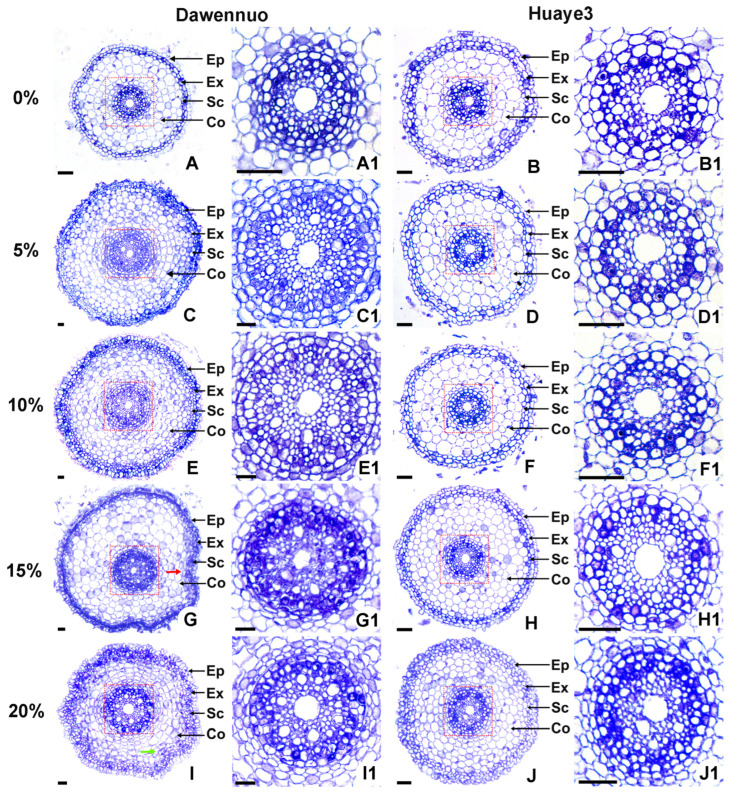
Comparison of transection of root tip elongation zone in Dawennuo and Huaye3 under different concentrations of PEG6000. (**A**,**C**,**E**,**G**,**I**) Root tip cells in Dawennuo under the different concentrations of PEG6000 treatment. (**A**) Transection of root tip elongation zone under the 0% PEG6000 treatment on the 7th day. (**C**) Transection of root tip elongation zone under the 5% PEG6000 treatment on the 7th day. (**E**) Transection of root tip elongation zone under the 10% PEG6000 treatment on the 7th day. (**G**) Transection of root tip elongation zone under the 15% PEG6000 treatment on the 7th day; The red arrow indicates the wrinkled cells in the apical elongation zone of Dawennuo. (**I**) Transection of root tip elongation zone under the 20% PEG6000 treatment on the 7th day; The green arrow indicates the wrinkled cells in the root tip elongation zone of Dawennuo. (**B**,**D**,**F**,**H**,**J**) Root tip cells in Huaye3 under the different concentrations of PEG6000 treatment. (**B**) Transection of root tip elongation zone under the 0% PEG6000 treatment on the 7th day. (**D**) Transection of root tip elongation zone under the 5% PEG6000 treatment on the 7th day. (**F**) Transection of root tip elongation zone under the 10% PEG6000 treatment on the 7th day. (**H**) Transection of root tip elongation zone under the 15% PEG6000 treatment on the 7th day. (**J**) Transection of root tip elongation zone under the 20% PEG6000 treatment on the 7th day. (**A1**–**J1**): Detailed diagram of the stele in the root tip elongation zone corresponding to red dotted boxes of (**A**–**J**). Bars = 100 μm in (**A**–**J**), Bars = 20 μm in (**A1**–**J1**). (Ep: Epidermis; Ex: Exodermis; Sc: Sclerenchyma; Co: Cortex).

**Figure 6 ijms-25-07134-f006:**
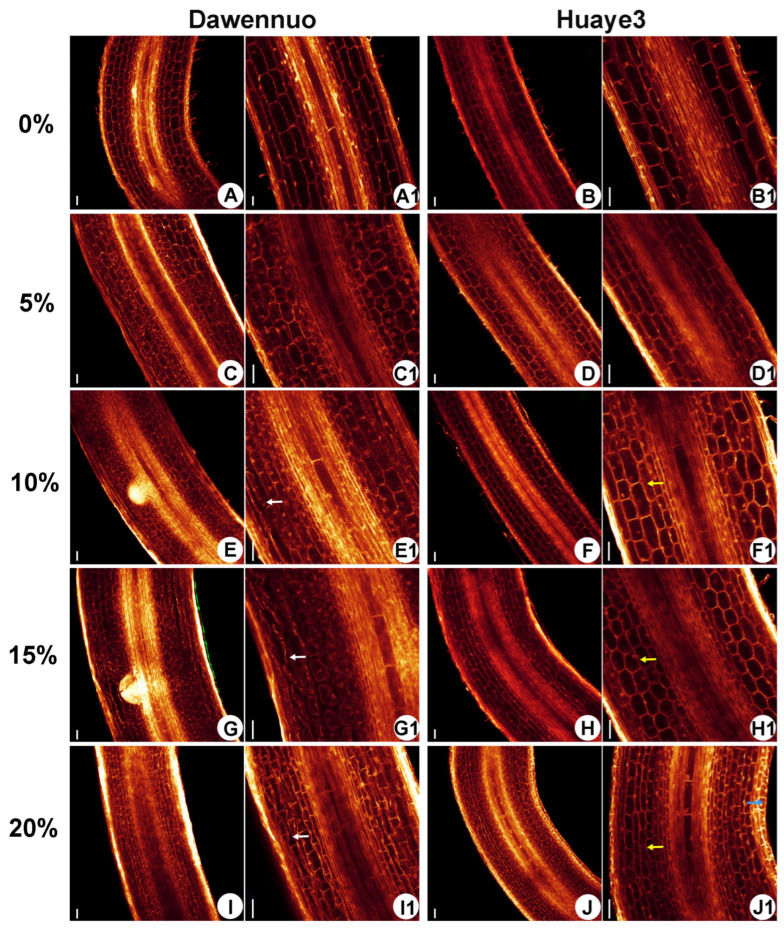
Comparison of morphological differences of root tip cells in Dawennuo and Huaye3 under the different concentrations of PEG6000 treatment. (**A**,**C**,**E**,**G**,**I**) Root tip cells in Dawennuo under the different concentrations of PEG6000 treatment. (**A**) Root tip cells with no treatment of PEG6000. (**C**) Root tip cells under the treatment of 5% PEG6000. (**E**) Root tip cells under the treatment of 10% PEG6000. (**G**) Root tip cells under the treatment of 15% PEG6000. (**I**) Root tip cells under the treatment of 20% PEG6000. (**B**,**D**,**F**,**H**,**J**) Root tip cells in Huaye3 under the different concentrations of PEG6000 treatment. (**B**) Root tip cells with no treatment of PEG6000. (**D**) Root tip cells under the treatment of 5% PEG6000. (**F**) Root tip cells under the treatment of 10% PEG6000. (**H**) Root tip cells under the treatment of 15% PEG6000. (**J**) Root tip cells under the treatment of 20% PEG6000. (**A1**–**J1**): Detailed diagram of the root tip elongation zone corresponding to (**A**–**J**). The white arrows indicate the deformed cells in the root tip elongation zone of Dawennuo appear deformed. The yellow arrows indicate the areas of increased cortex of the root tip elongation zone in Huaye3. Blue arrow indicates the epidermal cell thickening in the root tip elongation zone in Huaye3. Bars = 40 μm in (**A**–**J**,**A1**–**J1**).

**Figure 7 ijms-25-07134-f007:**
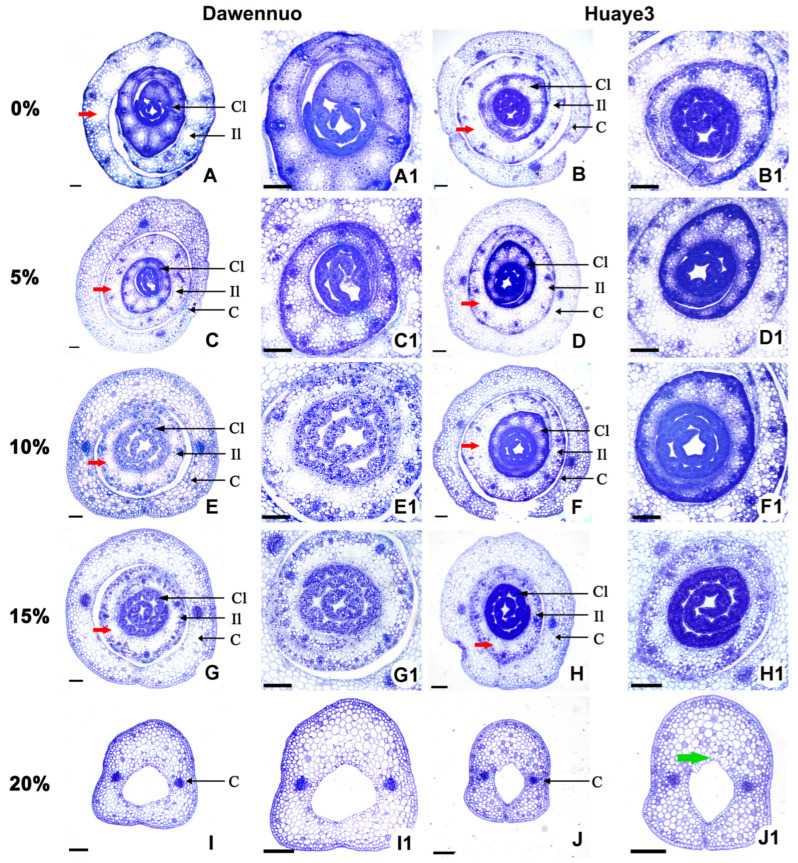
Comparison of morphological differences of bud cells in Dawennuo and Huaye3 under the different concentrations of PEG6000 treatment. (**A**,**C**,**E**,**G**,**I**) The bud cells of Dawennuo under the different concentrations of PEG6000 treatment. (**A**) The bud cells with no treatment of PEG6000. (**C**) The bud cells under the treatment of 5% PEG6000. (**E**) The bud cells under the treatment of 10% PEG6000. (**G**) The bud cells under the treatment of 15% PEG6000. (**I**) The bud cells under the treatment of 20% PEG6000. (**B**,**D**,**F**,**H**,**J**) The bud cells of Huaye3 under the different concentrations of PEG6000 treatment. (**B**) The bud cells with no treatment of PEG6000. (**D**) The bud cells under the treatment of 5% PEG6000. (**F**) The bud cells under the treatment of 10% PEG6000. (**H**) The bud cells under the treatment of 15% PEG6000. (**J**) The bud cells under the treatment of 20% PEG6000. (**A1**–**J1**): Detailed diagram of the developing cell structure of the bud corresponding to (**A**–**J**). Bars = 100 μm in (**A**–**J**,**A1**–**J1**). The red arrows indicate air cavity tissues. The green arrow indicates bulliform cells. Cl: Complete leaf; Il: Incomplete leaf; C: Coleoptile.

**Figure 8 ijms-25-07134-f008:**
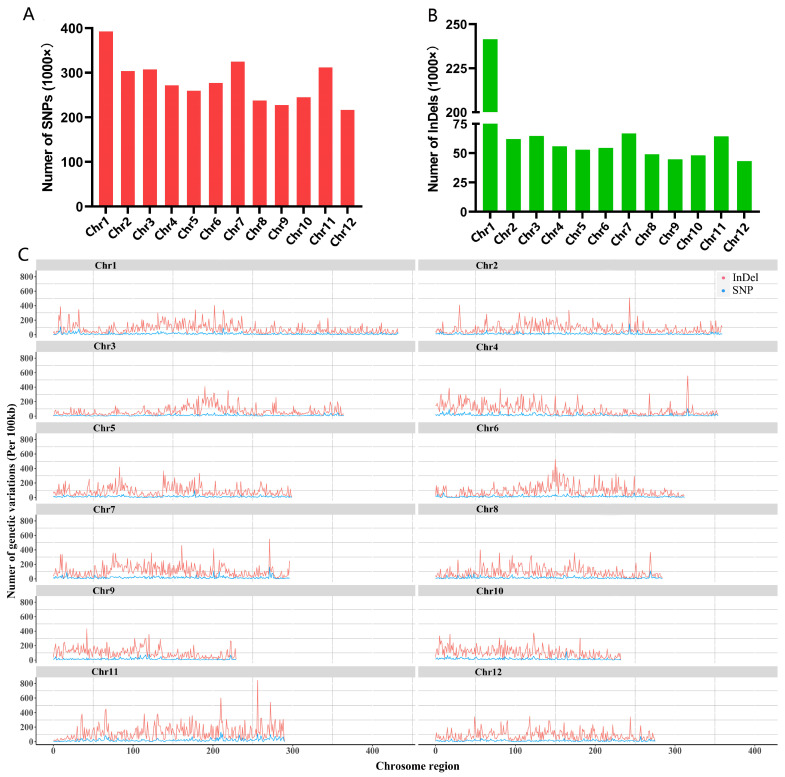
Genomic variation of Huaye3 and Dawennuo on each chromosome. (**A**) Number of SNP variations between Huaye3 and Dawennuo. (**B**) Number of InDels variation between Huaye3 and Dawennuo. (**C**) Distribution and variation of SNPs and InDels on each chromosome between Huaye3 and Dawennuo. The difference between SNPs (red) and InDels (blue) in Huaye3 is listed on the vertical axis, compared with Dawennuo. The positive numbers represent the change between Huaye3 and Dawennuo, and the negative numbers represent the differences between Huaye3 and Dawennuo.

**Figure 9 ijms-25-07134-f009:**
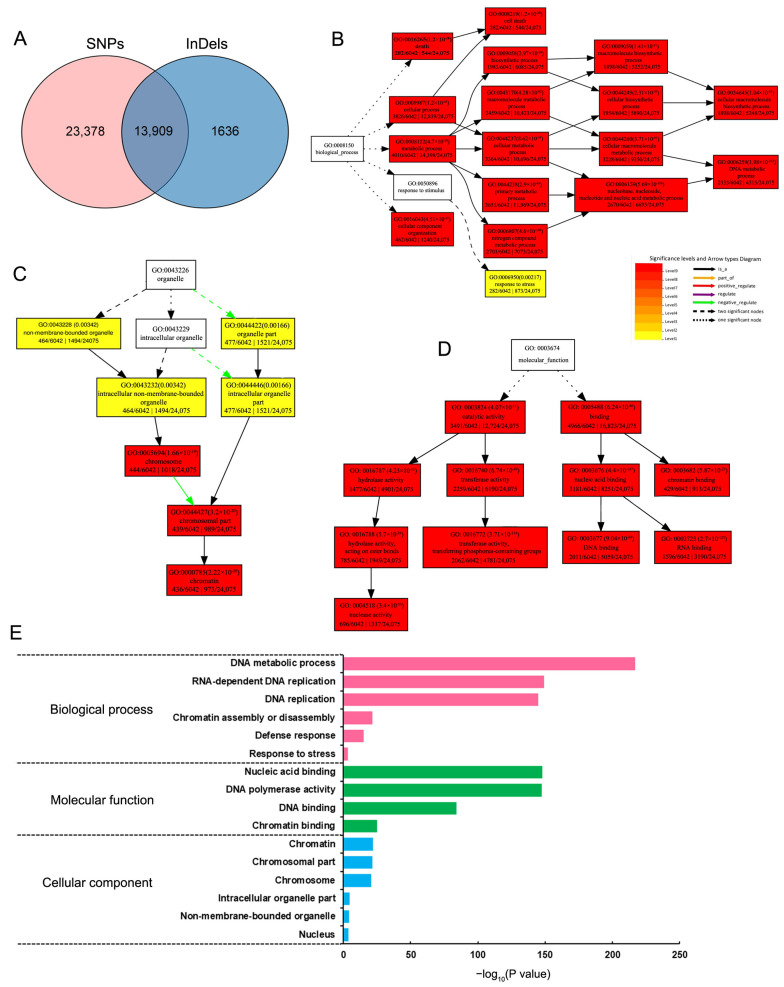
GO enrichment analysis of genomic variant genes in Huaye3 compared with Dawennuo. (**A**) Venn analysis of the differential SNPs and InDels in Huaye3 compared with Dawennuo. (**B**) GO enrichment analysis of biological process category in Huaye3 compared with Dawennuo. (**C**) GO enrichment analysis of molecular function category in Huaye3 compared with Dawennuo. (**D**) GO enrichment analysis of cellular component category in Huaye3 compared with Dawennuo. (**E**) Significant GO categories in Huaye3 compared with Dawennuo.

**Figure 10 ijms-25-07134-f010:**
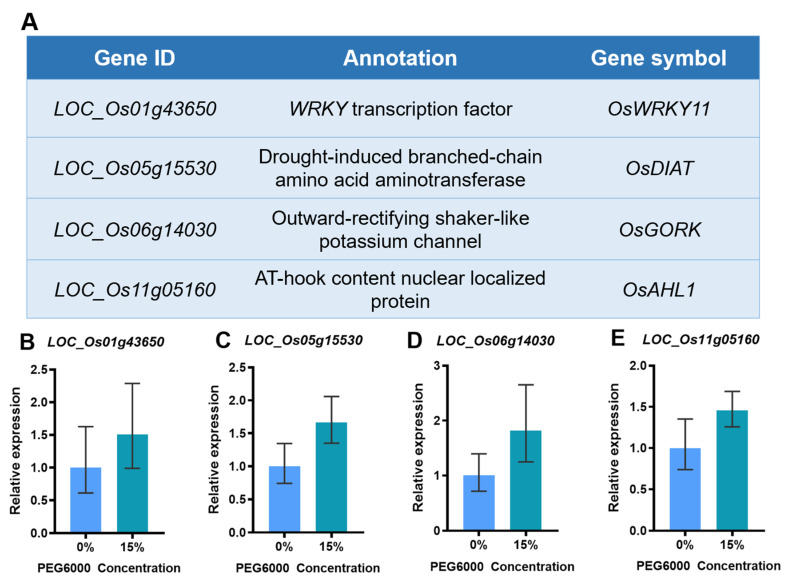
The qRT-PCR confirmation of drought-related genes in Huaye3 between 0% and 15%PEG6000. (**A**) Annotation of four drought-related genes identified from Huaye3 compared with Dawennuo. (**B**) The qRT-PCR confirmation of *LOC_Os01g43650* in Huaye3 between the 0% and 15% PEG6000. (**C**) The qRT-PCR confirmation of *LOC_Os05g15530* in Huaye3 between the 0% and 15% PEG6000. (**D**) The qRT-PCR confirmation of *LOC_Os06g14030* in Huaye3 between the 0% and 15% PEG6000 (**E**) The qRT-PCR confirmation of *LOC_Os11g05160* in Huaye3 between the 0% and 15% PEG6000.

## Data Availability

Data are contained within the article and Appendix A.

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
