# Peer review of "Comparative Cytological and Gene Expression Analysis Reveals That a Common Wild Rice Inbred Line Showed Stronger Drought Tolerance Compared with the Cultivar Rice"

_ijms, 2024, doi:10.3390/ijms25137134_

Round 1

Reviewer 1 Report

Comments and Suggestions for Authors
  1. In the abstract, include a comparison with Dawennuo, rather than just stating the drought resistance performance of Huaye3.
  2. Lines 146-147: The decreased fresh weight, such as 10.10 mg and 20.90 mg, is this per pot or per plant? Please provide a detailed description.
  3. Lines 135-136: The number of roots is not visible in Figure 3. It is recommended to add images of the root systems.
  4. Why is the absolute biomass of Huaye3 lower than Dawennuo at each drought concentration treatment in Figure 3? Wouldn't this indicate that Dawennuo is more drought-tolerant under severe drought stress?
  5. Why is the root-to-shoot ratio under 20% drought stress not shown in Figure 4? Figure 4A shows that the root-to-shoot ratio of Huaye3 is higher than Dawennuo under 20% drought stress. Why is there no difference under 15% drought stress, and what might be the reason for this?
  6. In Figure 5, the scale bars for A1, B1, D1, F1, G1, H1, J1 and C1, E1, G1, I1 are inconsistent, which might lead to inconsistent evaluation criteria.
  7. Why does Figure 5 show that the damage to Dawennuo root tip cell structure under 15% drought treatment is higher than under 20% drought treatment? Why does this occur?
  8. Discussion: Rice is one of the most researched crops in the world, with extensive studies on drought-resistant materials and their mechanisms. It is suggested to compare Huaye3 with other drought-resistant materials or varieties to explore the drought resistance characteristics of Huaye3.
  9. Lines 497-498: List the formulas for calculating the relative vigor index and drought tolerance index rather than just citing references.

Author Response

Response to Reviewer 1 Comments

Comments 1: In the abstract, include a comparison with Dawennuo, rather than just stating the drought resistance performance of Huaye3.

Response 1: Thanks for the kind suggestion. We have added the results of Dawennuo in the abstract.

Comments 2: Lines 146-147: The decreased fresh weight, such as 10.10 mg and 20.90 mg, is this per pot or per plant? Please provide a detailed description.

Response 2: Thanks for the kind suggestion. We have provided a detailed description of the fresh weight in the Lines of 146-147, and we also describe how to calculate the fresh weight in the methods section.

“The fresh weight of seedlings in Huaye3 and Dawennuo under different concentrations of PEG6000 were further analyzed. With the rice seedlings of Huaye3 and Dawennuo grown for up to 30 days, we found both the Huaye3 and Dawennuo were affected by the increase in PEG6000 concentration. Therefore, we selected five representative plants in each gradient treatment to evaluate the fresh weight of two materials. Compared with the no-treatment group, the fresh weight in Huaye 3 showed a slight effect on the drought stress treatment, decreasing by 10.10 mg and 20.90 mg under the 10% and 15% PEG6000 treatments, respectively (Fig. 3F). Compared with the no-treatment group in Dawennuo, the fresh weight was decreased by 18.30 mg, 50.50 mg, and 91.10 mg under the concentration treatment of 5%, 10%, and 15% PEG6000 (Figure 3F). The results indicated that Huaye3 showed less effect under the PEG6000 treatment than the Dawennuo”.

Comments 3: Lines 135-136: The number of roots is not visible in Figure 3. It is recommended to add images of the root systems.

Response 3: Thanks for the kind suggestion. In this study, the root tissue was treated with the PEG6000 solution, and it is hard to take photos of all rice seedlings together due to the PEG6000 solution is glued. Therefore, we provided the root tissue figure of each plant in Figure S3 to reveal the root difference of two materials under the PEG6000 treatment. The root results also showed the Huaye3 is less affected by the PEG6000.

Comments 4:Why is the absolute biomass of Huaye3 lower than Dawennuo at each drought concentration treatment in Figure 3? Wouldn't this indicate that Dawennuo is more drought-tolerant under severe drought stress?

Response 4: Thanks for the kind suggestion. The Huaye3 and Dawennuo belong to the different genetic backgrounds. Due to their distinct backgrounds, there are significant differences in phenotypic traits such as plant height, seed size, leaf width, and leaf length. The plant height of Huaye3 is much lower than the Dawennuo, and it looks like the absolute biomass of Huaye3 is lower than Dawennuo. To avoid the influence of the “absolute biomass” affected the drought tolerance of Huaye3 and Dawennuo, we used the no PEG treatment material as the control. Our results indicated that Dawennuo's fresh weight decreased by 18.3mg, 50.5mg, and 91.1mg under 5%, 10%, 15%, and 20% PEG6000 concentration treatments, respectively. In contrast, Huaye3 showed a reduction of 0mg, 10.1mg, and 20.9mg at the same PEG6000 concentrations. All of these results indicated that Dawennuo exhibited a significant decrease in fresh weight with increasing treatment concentrations compared to Huaye3, suggesting that Huaye3 is less affected by drought stress and possesses stronger drought tolerance.

Comments 5:Why is the root-to-shoot ratio under 20% drought stress not shown in Figure 4? Figure 4A shows that the root-to-shoot ratio of Huaye3 is higher than Dawennuo under 20% drought stress.

Response 5: Thanks for the kind suggestion. We have added the data of root-to-shoot ratio under 20% drought stress in the Figure 4. The root-to-shoot ratio of Huaye3 is higher than Dawennuo under 20% PEG treatment concentration.

Comments 6:Why is there no difference under 15% drought stress, and what might be the reason for this?

Response 6: Thanks for the kind suggestion. The Huaye3 and Dawennuo existed the difference, however, it was not arrived in significant difference.

Comments 7:In Figure 5, the scale bars for A1, B1, D1, F1, G1, H1, J1 and C1, E1, G1, I1 are inconsistent, which might lead to inconsistent evaluation criteria.

Response 7: Thanks for the kind suggestion. The Figure5 indicated the cytological difference of root tip elongation zone between the Dawennuo and Huaye3 under the PEG6000 treatment. Our results indicated that the root lengths were varied and exhibited the different length under the different concentrations of PEG6000. Therefore, the original sizes of root sections existed some variations, so the Figure5A-5J have inconsistent scale bars. The figure 5A1-5J1 is mainly focus on the structural and morphological of the cortical cells, so their scale bars are not inconsistent, but it can reflect the cytological difference.

Comments 8:Why does Figure 5 show that the damage to Dawennuo root tip cell structure under 15% drought treatment is higher than under 20% drought treatment? Why does this occur?

Response 8: We really appreciate the professional comment. It is an interesting phenomenon and need to study later. To make sure the damage degree of apical cell structure in Dawennuo under 15% PEG6000 concentration, we have already observed more than three different samples. We speculated that it may have some relationship with the drought response. The apical cells in Dawennuo under the 15% PEG6000 concentration may have no time to respond, so the cell structure is seriously damaged. With the PEG6000 concentration increased to 20%, the apical cells of Dawennuo became to adapt the drought stress. The epidermal cells are thickened and the number of cortical layers is increased to resist the effects of drought stress on the central column and ensure that the duct can actively and normally carry out water transport. These results is worth further studying the drought response and changes in rice root apical cells.

Comments 9:Discussion: Rice is one of the most researched crops in the world, with extensive studies on drought-resistant materials and their mechanisms. It is suggested to compare Huaye3 with other drought-resistant materials or varieties to explore the drought resistance characteristics of Huaye3.

Response 9: Thanks for the kind suggestion. We have added the comparison results of Huaye3 with other drought-resistant materials or varieties. The revised discussion was listed as follows:

“The root/shoot ratio is an important index for the drought tolerant ability. The root length, seedling length, and fresh weight were inhibited in varying concentrations of PEG6000 treatment. The root number of Huaye3 showed more stability and a slight effect on the fresh weight under the drought stress treatment when compared with the Dawennuo. These results were similar to the previous study. Two rice varieties, Swarnaprabha and Kattamodan, showed strong drought recovery ability, and their root/shoot ratio increased under the high concentration of PEG6000 [39]. In addition, the fresh weight in Huaye3 showed a slight effect on the drought stress treatment compared to the control group in Dawennuo. These results indicated that Huaye3 is more stable than the Dawennuo under drought stress treatment”.

Comments 10:Lines 497-498: List the formulas for calculating the relative vigor index and drought tolerance index rather than just citing references.

Response 10: Thanks for the suggestion. In the method section, we have listed the formulas for calculating the relative vigor index and drought tolerance index. The revision listed as follow:

 “The seed’s germination rate under the different treatments was investigated from the 3rd day. Germination index (GI) was calculated as= (Gt: Number of germinating seed grains on day tï¼›Dt:Corresponding days to germination)ï¼›Vigor index(VI)was calculated as=GI×S (S:Total length of roots and shoots of seeds); Relative vigor index(RVI)was calculated as= (VI of treatment group/ VI of the control group)×100%. Drought tolerance index was calculated as=GI of treatment group/ GI of the control group[48, 49]. All of these samples were performed on three replications”.

Reviewer 2 Report

Comments and Suggestions for Authors

The manuscript is prepared on a current topic and brings new knowledge in the given area. Before its acceptance, however, it is necessary to make certain additions and corrections, which I specify below in my review: 

Keywords - 50 % coincide with the title of the manuscript, therefore it is advisable to replace them.   

Introduction - is prepared at a very good level and appropriately represents the given issue on which the manuscript is focused.  

Results - the recorded results are described here in a suitable text form. I see the adjustment as a necessity, or addition of some Figures (2, 3, 4, and 10), where the legends are missing what is indicated by the check marks in the bar graphs. Yes, it is the variability of the given parameter, but it is expressed by what? It is similar to the line graph of Figure S2. This information should be in the legend of each Figure.  

Discussion - it is processed with high quality, but I would still dare to recommend the authors to include the current reference Khan et al.  (2024, DOI: 10.17221/88/2023-CJGPB) in section 3.2 or 3.3, which deals with abiotic stress and strigolactones, which can significantly influence drought resistance with their endogenous or exogenous activity. Among other things, the influence on the growth of rice roots and the influence of gene expression, which the authors annotated as a result of their study, are documented here.  

Material a Methods - for real-time PCR (line 537), the composition of the reaction (including the concentrations of individual components) and the temperature and time profile of the reaction are missing. This is crucial methodological information and is missing. Primer combinations are listed in Table S4. In the case of "Ubi", these are primers designed by the authors or taken from the literature. If they are taken over, the reference must be added. I would recommend to include the specific sizes of the products in the table, because in the text of the methodology, a range of 100-150 bp is indicated (which is really a lot for RT-qPCR). Has a primer effectiveness study been done? Or sequencing analyses of the obtained amplicons, which would prove that these are really PCR products for the given genes?

Formal remarks: 

In Table S1 - there are two different number formats (with a tangent and a comma).

In the References section, there are two different formats of references, i.e. one using the full name of the journals and the other using the abbreviated name of the journal. It is necessary to unify according to the Instruction for Authors.   

Based on the above, I recommend the manuscript for publication after major revision and second review.

Author Response

Comments 1: The manuscript is prepared on a current topic and brings new knowledge in the given area. Before its acceptance, however, it is necessary to make certain additions and corrections, which I specify below in my review.

Response 1: Thanks for the reviewer comments, we have revised the manuscript according to the reviewer’s comments and response it point-by-point.

Comments 2: Keywords - 50 % coincide with the title of the manuscript, therefore it is advisable to replace them. 

Response 2: We agree with the reviewer and have replaced the keywords using the new words. The revised Keywords were listed as follows: “Oryza rufipogon Griff; Drought stress; Seed germination; Genome re-sequencing”.

Comments 3: Introduction - is prepared at a very good level and appropriately represents the given issue on which the manuscript is focused.  

Response 3: We are very thankful to the reviewer for encouraging comments about the introduction section.

Comments 4: Results - the recorded results are described here in a suitable text form. I see the adjustment as a necessity, or addition of some Figures (2, 3, 4, and 10), where the legends are missing what is indicated by the check marks in the bar graphs. Yes, it is the variability of the given parameter, but it is expressed by what? It is similar to the line graph of Figure S2. This information should be in the legend of each Figure.

Response 4: Thanks for the kind suggestion. We have adjusted the legends of Figures (2, 3, 4, and 10) and also checked the other figures. For example, we revised the “treatment” to “PEG6000 concentration”, and listed all legends in the figures.

Comments 5: Discussion - it is processed with high quality, but I would still dare to recommend the authors to include the current reference Khan et al.  (2024, DOI: 10.17221/88/2023-CJGPB) in section 3.2 or 3.3, which deals with abiotic stress and strigolactones, which can significantly influence drought resistance with their endogenous or exogenous activity. Among other things, the influence on the growth of rice roots and the influence of gene expression, which the authors annotated as a result of their study, are documented here.  

Response 5: Thanks for the kind suggestion. We have added the relevant literature in the discussion section. The revision is listed as follows:

“Drought stress is a critical environment limiting plant growth and productivity [40]. Root tissue is generally responsive to the drought-stress environment. The phytohormones, such as the Strigolactones (SLs), regulated the root system and responsive to the drought stress treatment [41]”.

Comments 6: Material a Methods - for real-time PCR (line 537), the composition of the reaction (including the concentrations of individual components) and the temperature and time profile of the reaction are missing. This is crucial methodological information and is missing.

Response 6: Thanks for the kind suggestion. We have revised the methods of real-time PCR. We added the composition of the reaction and the temperature and time profile of the reaction in this study. The revision is listed as follows:

“The RT-qPCR experiment was performed on the Roche Lightcycler480 instrument with the final volumes of 20 μL, each containing 1 μL of cDNA, 0.4 μL of each primer (10 μM), 10 μL 2× Hieff qPCR SYBR Green Master Mix (No Rox), and 8.2 μL ddH2O. The reaction conditions were as follows: 30 s at 95 °C, 40 cycles of 95 °C denaturation for 10 s, and 60 °C annealing and extension for 30 s. The rice ubiquitin gene was used as the reference gene, and its sequence is listed in Table S5. After the reaction data was obtained, the gene expression level was calculated using the 2-△△Ct method [59]. The RT-qPCR for each gene had three technical replicates and three biological replicates”.

Comments 7: Primer combinations are listed in Table S4. In the case of "Ubi", these are primers designed by the authors or taken from the literature. If they are taken over, the reference must be added.

Response 7: Thanks for the kind suggestion. We have listed the primer sequence and product size of "Ubi" reference gene in Table S5.

Comments 8: I would recommend to include the specific sizes of the products in the table, because in the text of the methodology, a range of 100-150 bp is indicated (which is really a lot for RT-qPCR).

Response 8: Thanks for the kind suggestion. We have listed the product size of primers in Table S5.

Comments 9: Has a primer effectiveness study been done? Or sequencing analyses of the obtained amplicons, which would prove that these are really PCR products for the given genes?

Response 9: Yes. We have tested the primers before the expression analysis. We used the agarose gel electrophoresis and melting curve to check the specificity of the primers before the RT-qPCR experiment. In the revision, we provided the melting curve in FigureS4 to verify the effectiveness of primer.

Comments 10: In Table S1 - there are two different number formats (with a tangent and a comma).

Response 10: Yes. We revised Table S1 and made it in the same format.

Comments 11: In the References section, there are two different formats of references, i.e. one using the full name of the journals and the other using the abbreviated name of the journal. It is necessary to unify according to the Instruction for Authors.   

Response 11: Thanks for the suggestion. We have corrected the references' formats according to the Instructions for Authors.

Comments 12: Based on the above, I recommend the manuscript for publication after major revision and second review.

Response 12:  We agree with the reviewer and have revised and corrected the text throughout the manuscript.

Round 2

Reviewer 2 Report

Comments and Suggestions for Authors

I don't know why we gained access in the MDPI system to the authors' responses to another reviewer's review. Yes, in some journals, reviewers receive answers to all reviewers' questions.   

I tried to find all my comments in the edited text and I can only state that the adjustments have been partially made. I specify my requirements below:   

line 55-58 - I recommend adding a reference that deals with the study of local rice genotypes for drought using PEG (i.e. the subject of the manuscript) - Rahim et al. (2020 - DOI: 10.3390/app10134471);   

Fig. 10 and Fig. S2 - the legend does not indicate what characterizes the individual line segments for the given parameter value (is this a standard error?, if so, it must be stated);   

Table S4 - I asked for information on primers (v pÅ™edchozím review), but given that I have the answers to another review available, I cannot evaluate the answer, but the adjustment was not made;   

Fig. S3 and S4 - new references to these supplementary materials appear in the text, but they are not part of the submission.   

Based on these facts, I recommend the manuscript for publication after minor revision, because I assume that there must have been some mistake that will be explained.

Author Response

Response to Reviewer 2 Comments in Round 2

Comments and Suggestions for Authors

I don't know why we gained access in the MDPI system to the authors' responses to another reviewer's review. Yes, in some journals, reviewers receive answers to all reviewers' questions.   

Comments 1: I tried to find all my comments in the edited text and I can only state that the adjustments have been partially made.

    Response 1: Thanks very much. We have carefully revised the manuscript according to the two-round comments. To clarify our results, we highlight the revised contents in red according to the reviewer in the first round and in blue according to the reviewer in the second round.

In addition, we also provided the point-to-point response to comments in the first round at the end of the comments to ensure that all adjustments were made.

Comments 2: Line 55-58 - I recommend adding a reference that deals with the study of local rice genotypes for drought using PEG (i.e. the subject of the manuscript) - Rahim et al. (2020 - DOI: 10.3390/app10134471); 

    Response 2: Thank you for your suggestion. The paragraph of Lines 55-58 is mainly about the advantages of common wild rice. After careful consideration, we think this reference is more important for our discussion, so we added it to the discussion section (lines 413-415).

Comments 3: Fig. 10 and Fig. S2 - the legend does not indicate what characterizes the individual line segments for the given parameter value (is this a standard error?, if so, it must be stated);   

    Response 3: Thanks for the kind suggestion. We have revised Fig.10 and Fig.S2, respectively, and inserted them in the manuscript. We also added a description of individual line segments, which refers to standard errors.

Comments 4: Table S4 - I asked for information on primers (v pÅ™edchozím review), but given that I have the answers to another review available, I cannot evaluate the answer, but the adjustment was not made;   

    Response 4: Sorry, it is a mistake. The “Table S4” of the methods section should be revised as Table S5. Table S5 provides information on the product size of all primers.

Comments 5: Fig. S3 and S4 - new references to these supplementary materials appear in the text, but they are not part of the submission.   

    Response 5: Thanks for the suggestion. I think we did not clarify the reasons for adding the additional figures (Fig. S3 and S4) and table (Table S2) in the first revised version (R1). In the review of Round 1, reviewer 1 wanted us to provide information about primers' effectiveness, so we added Fig. S4 to the supplemental file. Fig. S4 shows the melting curve of all primers and can reflect the specificity of primers.

    Reviewer 2 wants us to provide the exact data and root tissue figures of Dawennuo and Huaye3 under the different concentrations of PEG6000. Therefore, we have provided the additional Figure (Fig. S3) and Table (Table S2) in the R1. Both figures are essential to the revision. Fig. S3 indicated the difference in root tissue in Dawennuo and Huaye3 under the PEG6000 treatment. Table S2 provides the exact root and shot tissue data in the seeding stage.

Comments 6: Based on these facts, I recommend the manuscript for publication after minor revision, because I assume that there must have been some mistake that will be explained.

Response 6: We agree with the reviewer. We have revised and corrected the text throughout the manuscript.

Response to Reviewer 2 Comments in Round 1

Point-by-point response to Comments and Suggestions for Authors

Comments 1: In the abstract, include a comparison with Dawennuo, rather than just stating the drought resistance performance of Huaye3.

Response 1: Thanks for the kind suggestion. We have added the results of Dawennuo in the abstract.

Comments 2: Lines 146-147: The decreased fresh weight, such as 10.10 mg and 20.90 mg, is this per pot or per plant? Please provide a detailed description.

Response 2: Thanks for the kind suggestion. We have provided a detailed description of the fresh weight in the Lines of 146-147, and we also describe how to calculate the fresh weight in the methods section.

“The fresh weight of seedlings in Huaye3 and Dawennuo under different concentrations of PEG6000 were further analyzed. With the rice seedlings of Huaye3 and Dawennuo grown for up to 30 days, we found both the Huaye3 and Dawennuo were affected by the increase in PEG6000 concentration. Therefore, we selected five representative plants in each gradient treatment to evaluate the fresh weight of two materials. Compared with the no-treatment group, the fresh weight in Huaye 3 showed a slight effect on the drought stress treatment, decreasing by 10.10 mg and 20.90 mg under the 10% and 15% PEG6000 treatments, respectively (Fig. 3F). Compared with the no-treatment group in Dawennuo, the fresh weight was decreased by 18.30 mg, 50.50 mg, and 91.10 mg under the concentration treatment of 5%, 10%, and 15% PEG6000 (Figure 3F). The results indicated that Huaye3 showed less effect under the PEG6000 treatment than the Dawennuo”.

Comments 3: Lines 135-136: The number of roots is not visible in Figure 3. It is recommended to add images of the root systems.

Response 3: Thanks for the kind suggestion. In this study, the root tissue was treated with the PEG6000 solution, and it is hard to take photos of all rice seedlings together due to the PEG6000 solution is glued. Therefore, we provided the root tissue figure of each plant in Figure S3 to reveal the root difference of two materials under the PEG6000 treatment. The root results also showed the Huaye3 is less affected by the PEG6000.

Comments 4:Why is the absolute biomass of Huaye3 lower than Dawennuo at each drought concentration treatment in Figure 3? Wouldn't this indicate that Dawennuo is more drought-tolerant under severe drought stress?

Response 4: Thanks for the kind suggestion. The Huaye3 and Dawennuo belong to the different genetic backgrounds. Due to their distinct backgrounds, there are significant differences in phenotypic traits such as plant height, seed size, leaf width, and leaf length. The plant height of Huaye3 is much lower than the Dawennuo, and it looks like the absolute biomass of Huaye3 is lower than Dawennuo. To avoid the influence of the “absolute biomass” affected the drought tolerance of Huaye3 and Dawennuo, we used the no PEG treatment material as the control. Our results indicated that Dawennuo's fresh weight decreased by 18.3mg, 50.5mg, and 91.1mg under 5%, 10%, 15%, and 20% PEG6000 concentration treatments, respectively. In contrast, Huaye3 showed a reduction of 0mg, 10.1mg, and 20.9mg at the same PEG6000 concentrations. All of these results indicated that Dawennuo exhibited a significant decrease in fresh weight with increasing treatment concentrations compared to Huaye3, suggesting that Huaye3 is less affected by drought stress and possesses stronger drought tolerance.

Comments 5:Why is the root-to-shoot ratio under 20% drought stress not shown in Figure 4? Figure 4A shows that the root-to-shoot ratio of Huaye3 is higher than Dawennuo under 20% drought stress.

Response 5: Thanks for the kind suggestion. We have added the data of root-to-shoot ratio under 20% drought stress in the Figure 4. The root-to-shoot ratio of Huaye3 is higher than Dawennuo under 20% PEG treatment concentration.

Comments 6:Why is there no difference under 15% drought stress, and what might be the reason for this?

Response 6: Thanks for the kind suggestion. The Huaye3 and Dawennuo existed the difference, however, it was not arrived in significant difference.

Comments 7:In Figure 5, the scale bars for A1, B1, D1, F1, G1, H1, J1 and C1, E1, G1, I1 are inconsistent, which might lead to inconsistent evaluation criteria.

Response 7: Thanks for the kind suggestion. The Figure5 indicated the cytological difference of root tip elongation zone between the Dawennuo and Huaye3 under the PEG6000 treatment. Our results indicated that the root lengths were varied and exhibited the different length under the different concentrations of PEG6000. Therefore, the original sizes of root sections existed some variations, so the Figure5A-5J have inconsistent scale bars. The figure 5A1-5J1 is mainly focus on the structural and morphological of the cortical cells, so their scale bars are not inconsistent, but it can reflect the cytological difference.

Comments 8:Why does Figure 5 show that the damage to Dawennuo root tip cell structure under 15% drought treatment is higher than under 20% drought treatment? Why does this occur?

Response 8: We really appreciate the professional comment. It is an interesting phenomenon and need to study later. To make sure the damage degree of apical cell structure in Dawennuo under 15% PEG6000 concentration, we have already observed more than three different samples. We speculated that it may have some relationship with the drought response. The apical cells in Dawennuo under the 15% PEG6000 concentration may have no time to respond, so the cell structure is seriously damaged. With the PEG6000 concentration increased to 20%, the apical cells of Dawennuo became to adapt the drought stress. The epidermal cells are thickened and the number of cortical layers is increased to resist the effects of drought stress on the central column and ensure that the duct can actively and normally carry out water transport. These results is worth further studying the drought response and changes in rice root apical cells.

Comments 9:Discussion: Rice is one of the most researched crops in the world, with extensive studies on drought-resistant materials and their mechanisms. It is suggested to compare Huaye3 with other drought-resistant materials or varieties to explore the drought resistance characteristics of Huaye3.

Response 9: Thanks for the kind suggestion. We have added the comparison results of Huaye3 with other drought-resistant materials or varieties. The revised discussion was listed as follows:

“The root/shoot ratio is an important index for the drought tolerant ability. The root length, seedling length, and fresh weight were inhibited in varying concentrations of PEG6000 treatment. The root number of Huaye3 showed more stability and a slight effect on the fresh weight under the drought stress treatment when compared with the Dawennuo. These results were similar to the previous study. Two rice varieties, Swarnaprabha and Kattamodan, showed strong drought recovery ability, and their root/shoot ratio increased under the high concentration of PEG6000 [39]. In addition, the fresh weight in Huaye3 showed a slight effect on the drought stress treatment compared to the control group in Dawennuo. These results indicated that Huaye3 is more stable than the Dawennuo under drought stress treatment”.

Comments 10:Lines 497-498: List the formulas for calculating the relative vigor index and drought tolerance index rather than just citing references.

Response 10: Thanks for the suggestion. In the method section, we have listed the formulas for calculating the relative vigor index and drought tolerance index. The revision listed as follow:

 “The seed’s germination rate under the different treatments was investigated from the 3rd day. Germination index (GI) was calculated as=  (Gt: Number of germinating seed grains on day tï¼›Dt:Corresponding days to germination)ï¼›Vigor index(VI)was calculated as=GI×S (S:Total length of roots and shoots of seeds); Relative vigor index(RVI)was calculated as= (VI of treatment group/ VI of the control group)×100%. Drought tolerance index was calculated as=GI of treatment group/ GI of the control group[48, 49]. All of these samples were performed on three replications”.

Round 3

Reviewer 2 Report

Comments and Suggestions for Authors

The authors accepted all my comments and I recommend the manuscript for publication.